# Super-Alarms with Diagnosis Proficiency Used as an Additional Layer of Protection Applied to an Oil Transport System

**DOI:** 10.3390/e23020139

**Published:** 2021-01-23

**Authors:** John W. Vásquez, Gustavo Pérez-Zuñiga, Javier Sotomayor-Moriano, Adalberto Ospino

**Affiliations:** 1Research Group GPS, Universidad de Investigación y Desarrollo—UDI, Bucaramanga 680004, Colombia; 2Departamento de Ingeniería, Pontificia Universidad Católica del Perú—PUCP, Avenida Universitaria 1801, San Miguel, Lima 15088, Peru; gustavo.perez@pucp.pe (G.P.-Z.); jsotom@pucp.edu.pe (J.S.-M.); 3Research Group GIOPEN, Universidad de la Costa—CUC, Barranquilla 080014, Colombia; aospino8@cuc.edu.co

**Keywords:** alarm management, protection layers, safe-process, Super-Alarm, diagnosis

## Abstract

In automated plants, particularly in the petrochemical, energy, and chemical industries, the combined management of all of the incidents that can produce a catastrophic accident is required. In order to do this, an alarm management methodology can be formulated as a discrete event sequence recognition problem, in which time patterns are used to identify the safe condition of the process, especially in the start-up and shutdown stages. In this paper, a new layer of protection (a Super-Alarm), based on the diagnostic stage to industrial processes is presented. The alarms and actions of the standard operating procedures are considered to be discrete events involved in sequences; the diagnostic stage corresponds to the recognition of the situation when these sequences occur. This provides operators with pertinent information about the normal or abnormal situations induced by the flow of the alarms. Chronicles Based Alarm Management (CBAM) is the methodology used in this document to build the chronicles that will permit us to generate the Super-Alarms; in addition, a case study of the petrochemical sector using CBAM is presented in order to build one chronicle that represents the scenario of an abnormal start-up of an oil transport system. Finally, the scenario’s validation for this case is performed, showing the way in which, a Super-Alarm is generated.

## 1. Introduction

Today, the expanding complexity of control systems is due to the increasing automation of industrial production processes. The use of digital information-based technologies in these systems suggests an increase in the amount of data that must be monitored and processed, including better communication ability between the agents of the process [1]. The automatic reconfiguration of embedded control systems is a usual requirement for highly automated systems, and the applications of fault diagnosis are difficult to implement [2,3]; consequently, the ultimate goal for a supervision and control system is to optimize the availability, reliability, and safety of production processes [4]. With regards to safety, the integrated management of the critical factors in the process ensures an optimum reliability level in the industrial plants [5,6]. Factors such as the control of the process variables, procedures, and steps followed in the transitional stages are intended to keep the plants within the operating established limits [7,8]. On the starting or shutdown procedures, the quantity of signals increases, so the plant’s safety needs to involve the integrated management of those factors when analyzing the causes of the accidents. In other words, these factors must be managed together, and not separately, because if any of them is left outside, unattended or decreased, the security would be threatened [9,10]. When one industrial process changes its state, for example, its start-up and shutdown stages, the alarm flood spreads, and it causes severe situations in which the operator cannot react correctly. Besides this, it is commonly reported that 70% of plant conflicts happen at the start-up/shutdown stages [11]. Due to this alarm flood, dynamic alarm management is needed. Nowadays, many fault detection and diagnosis methods for multimode processes have been proposed; however, these techniques cannot register fundamental faults in the basic alarm system [12]. Consequently, the operators need a tool that helps them to recognize the plant’s situation, especially in the transitional stages such as start-up and shutdown.

Safety conditions and the advancing performance in the monitoring, control, and management of complex systems have stimulated notable interest and efforts dedicated to the advancement of fault detection and isolation techniques. This raises the need not only for a diagnosis system that helps to maintain the safety increasing the availability of the installation, but also for new alarm management methodologies [13]. Industrial plant safety involves the integrated management of all of the factors that may cause accidents. As such, alarm management is one aspect of great interest in safety planning for different plants. Any additional support in the protection of industrial processes will be well received in the process of safeprocess community. This article is divided into four sections. Section 1 presents the introduction. Section 2 describes the research method, including the traditional layers of protection in an industrial process and the Super-Alarm as a new layer of protection; furthermore, the Chronicle Based Alarm Management methodology is also presented in this section. Section 3 presents a case study with the results analysis. Section 4 corresponds to the conclusions.

## 2. Research Method 

This section presents a research method which includes the proposal of a new protection layer, called a Super-Alarm, followed by the description of the Chronicle Based Alarm Management methodology used to generate Super-Alarms, which will help in the diagnosis of the industrial processes, especially in the startup and shutdown stages.

### 2.1. Layers of Protection and the Super-Alarm Layer

The operation of many industrial processes, especially in the chemical, mineral, energy and petrochemical sectors, involves inherent risks due to the presence of dangerous materials like gases and chemicals, which in some conditions can cause emergencies. In these types of industrial processes, safety is supplied by layers of protection [14], which begin with a safe design (the Process design level) and an effective process control (the Process Control level), followed by the manual (the Operator interventions level) and automatic (the Safety Instrumented System level) prevention layers, and concluding with layers to mitigate the consequences of a critical event (the Active protection level, Passive protection level, Plant emergency response level, and Community emergency response level), as shown in Figure 1.

Diagnosis in industrial processes corresponds to the procedures, activities, and tools that help operators to recognize the real plant situation, especially at transitional stages in which the risk of accidents increases. Figure 2 presents the process safety relationships, in which (at the left of the figure) the protection layers (Loop, Alarm, and Trip) are related to all of the elements of the supervision scheme. With regard to the components of the supervision scheme, the first level includes the instrumentation and actuators of the system, including the Safety Instrumented System (SIS). The next level contains the acquisition and control equipment, followed by the supervision stage, in which the tools of diagnosis are implemented. Now, these tools of diagnosis could be a new protection layer in the process if it gives relevant information to the operators, especially when an alarms flood occurs. The goal of supervision and control tools is to maintain the process variables between its limits of operation.

In order to determine the events and signals of a procedure, it is necessary to analyze and consider the initial conditions of the process, and to identify possible failure modes. Hence, a complex system requires a division into subsystems to allow a reliable analysis. The goal of the technology used is to maintain the process variables on their limits of operation. One additional layer of protection could reduce the accident probability, helping the operators to take better decisions when alarm floods happen. It has been demonstrated that advanced diagnostic systems for industrial processes, together with the interventions of the operators, may constitute an additional protective safety layer [15]. However, these new elements seem to have never been included as a layer of protection because diagnostic systems for industrial processes are not yet extensive in practical tools [16]. In terms of process safety, the principal characteristics of a good protective barrier are specificity, independence, reliability, and audit. ‘Specificity’ refers to a barrier that is capable of detecting and preventing or mitigating the consequences of a potentially dangerous specific event (e.g., explosion). ‘Independence’ refers to a barrier which is independent of all of the other layers which are associated with the potentially dangerous event, when there is no potential for common cause failures. Furthermore, the protection layer is independent of the initiating event. ‘Reliability’ refers to the protection provided by the barrier, which reduces the risk identified for a specific and known quantity, which is then determined by its probability of failure. ‘Auditing’ refers to the fact that a barrier must be designed to allow inspections, and the periodic and regular testing of the protection function [17,18].

A new protection barrier called a Super-Alarm has been proposed in [19,20], situated between the layer Alarm and the layer Trip (SIS); see Figure 3. This new barrier comes from a diagnosis process, and it is specific because it is capable of detecting and preventing one specific (particular) dangerous situation, e.g., the wrong operative action in the start-up procedure, or a failure in one valve. This new barrier is independent because its functionality does not depend on the other elements: if some of the signals involved in the diagnosis tool fail, this new tool could detect it. The reliability of this barrier is determined by the reduction of the large number of alarms avoided by the operators. Finally, this new protection layer can be audited, because the diagnosis tools permit its revision from a methodology that includes simulations of scenarios checking the response. The concept of a Super-Alarm corresponds to a new alert to the operators resulting from a diagnosis procedure representing a superior alarm. Consequently, in automatic control systems, the supervision functions serve to indicate undesirable or unpermitted process states, and takes appropriate actions that maintain performance and avoid damage or harm states. A system is said to be diagnosable if whatever the behavior of the system, it will be able to determine, without ambiguity, a unique diagnosis. When a super-alarm is generated, the supervision and control system can provoke automatic control actions in addition to the alerts to the operators. The diagnosability of a system is generally computed from its model [21]; in applications using a model-based diagnosis, such a model is already present and does not need to be built from scratch. The methodology used to generate super-alarms in this paper is supported by an event-based diagnosis process in which, from a flow of discrete events, normal and abnormal situations can be detected. The fault diagnosis in general consists in the following three important aspects: ‘fault detection’ consists in discovering the existence of faults in the most useful units in the process; ‘fault isolation’ refers to the localization (classification) of the different faults; ‘fault analysis or identification’ consists in determining the type, degree and origin of the fault [22]. In this paper, a fault is considered to be the consequence of a sequence of discrete events that represent this faulty scenario; a fault is not considered to be a single fault event. In conclusion, a super-alarm corresponds to a new element resulting from a diagnosis process in which risk and hazard analysis are required. Designing and constructing Super-Alarms in a supervisory system requires a methodology that gives us relevant information about the process according to the events and procedural actions that have occurred.

### 2.2. Chronicle Based Alarm Management Methodology

**Definition** **1.**
*An event e is defined as a pair e = (*
$\sigma_{i},\;t_{i}$
*), in which *
$\sigma_{i}$
*∈ E is an event type, and*
$t_{i}$
*is a variable of an integer type called the event date. E correspond to the set of the totally event types of the system. Several events can have the same type of event, but do not necessarily have the same date; for instance e*_1_* = (a, 3) and e*_2_* = (a, 6) are two events that carry the same type of event (a).*


A flow of activity generated by a system is represented by a temporal sequence. In these temporal sequences, the time is represented by a discrete set of time points which is totally ordered, and whose granularity is sufficiently thin compared to the observed dynamics; given the precision permitted by the means of observation, we can assume that there is no inaccuracy. In the following, we may refer to an event type as an event for brevity. A temporal sequence (or a sequence, for short) consists of several events which take place in an orderly manner, which leads us to the following definition:

**Definition** **2.**
*A sequence on E is denoted as an ordered set of events S =*
${\left(\sigma_{i},\;t_{i}\right)}_{j}$
*with j ∈ N_l_, in which l is the size of the temporal sequence S, and N_l_ is a finite set of linearly ordered instants of cardinality l. Furthermore, l = |S| is the size of the temporal sequence, i.e., the number of event type occurrences in S. An example of a sequence representing an activity stream may be given by the sequence*
$S_{1}=\left\{e_{1},e_{2},e_{3},e_{4},e_{5},e_{6}\right\}=\left\{\left(a,2\right),\left(b,4\right),\left(c,5\right),\left(a,8\right),\left(b,9\right),\left(a,10\right)\right\}$
*with l*_1_* = 6.*


**Definition** **3.**
*A chronicle is defined as a triplet C = 〈ξ,Τ,G〉 [23], such that: ξ ⊆ E, in which ξ is called the typology of the chronicle, and Τ is the set of temporal constraints of the chronicle. G = (Ψ,A) is a directed graph in which:*
Ψ is a set of indexed event types, i.e., a finite indexed family defined by ψ: H → E, in which H ⊏ N.A is a set of edges between the indexed event types; there is an edge ($\sigma_{1\left(h1\right)}$, $\sigma_{2\left(h2\right)}$) ∈ A if and only if there is a time constraint between $\sigma_{1\left(h1\right)}$, and $\sigma_{2\left(h2\right)}$.


**Definition** **4.***The chronicle instance: a chronicle C = 〈 ξ,Τ,G 〉 is recognized in a temporal sequence S of event types ξ´, such that ξ ⊆ ξ´, when all temporal constraints Τ are satisfied. Then, C_inst_ = 〈 ξ´, Τ_v_ 〉 in which Τ_v_ is a valuation of Τ. If the sequence S has finished, and at least one event that occurs violates some temporal constraint, this chronicle is not recognized. Figure 4 illustrates the above definition: the chronicle on the left is recognized in the first and second sequence. Nevertheless, it is not recognized in the third sequence, because the only set of constraints relating a,b,c, and d in this sequence (Sequence_3_) is: Τ_v_ = *{*a*[5,5]*b*; *a*[3,3]*c*; *c*[2,2]*b*; *b*[2,2]*d*}*, and Τ_v_ is not a valuation of T =* {*a*[3,4]*b*; *a*[1,2]*c*; *c*[1,2]*b*; *b*[1,2]*d*}.

**Definition** **5.***The temporal restriction: a temporal restriction for a pair of event types* ($\sigma_{i}$*,*
$\sigma_{j}$) *is a given time constraint between their event dates*
$TR_{ij}=\sigma_{i}\left[t^{-},t^{+}\right]\sigma_{j}$.

The principle of Chronicle Based Alarm Management (CBAM) is to consider several process situations (normal or abnormal) during the start-up and shutdown stages, and to model each one of these situations through a learned chronicle. For this, given the situation to be modeled, the algorithm HCDAM (Heuristic Chronicle Discovery Algorithm Modified) is fed by a set of event sequences that are structured from simulations and the expert knowledge, giving us the respective chronicle of each situation [24]. Finally, when these chronicles are recognized, a Super-Alarm can be generated, giving relevant information to the operator’s, and we can assume that it as a new layer of protection from which actions can reduce the accident occurrences because, in many situations of alarm flood, hazardous scenarios happen. The global objective of CBAM is to generate a chronicle database on which a diagnosis process based on chronicle recognition is then performed. This new methodology relies then on three main steps, as shown below:

STEP 1: Event type identification. The aim of this step is to determine the event types that define the chronicles. For this step, information from the standard operating procedures and from the evolution of the continuous variables is exploited.

STEP 2: Event sequence generation. From the expertise and an event abstraction procedure, this step determines the date of occurrence of each event type for the construction of the representative event sequences used by a learning algorithm. A representative event sequence is the set of event types with their dates of occurrence that can be associated with a specific scenario of the process. The representative event sequences are then verified using the hybrid modeling of the system and the hybrid causal graphs.

STEP 3: Chronicle database construction. For each scenario, the representative event sequences and temporal restrictions are given by experts, and these elements are taken to learn chronicles. In order to learn chronicles, this step uses the extended version of the Heuristic Chronicle Discovery Algorithm (HCDAM), which is described in [10,22]. The set of chronicles learned for each scenario and each process element constitutes the chronicle database. A complex process *Pr* is composed of different units or areas *Pr* = {*Ar*_1_, *Ar*_2_, *Ar*_3_, ……. *Ar*_n_} in which each area has φ operational modes (e.g start-up, shutdown, slow march, etc.) noted *O_i_*, *i* = 1,2,3...φ. The process behavior in each operating mode can be either normal or faulty. The set of failure labels is defined as Δ*_f_* = {*f*_1_, *f*_2_, *f*_3_, …. *f*_r_}, and the complete set of possible labels is $\Delta =N⋃\Delta_{f}$, in which *N* means normal. In order to monitor the process and to recognize the different situations (normal or faulty) of the operational modes, it is proposed to build a chronicle base for each area. For a given area, a learned chronicle $C_{ij}^{m}$ is associated with each couple ($O_{i},\;L_{j}$) in which $L_{j}\in \Delta$. Equation (1) determines the set of chronicles *C* for any process area ($Ar_{m}$).
(1)$$CAr_{m}=\begin{array}{c} O_{1}\\ \begin{array}{c} O_{2}\\ \ldots \\ O_{k} \end{array} \end{array}\left|\begin{array}{cc} N & \begin{array}{cc} f_{1} & \begin{array}{cc} f_{2} & \begin{array}{cc} \ldots & f_{r} \end{array} \end{array} \end{array}\\ \begin{array}{c} C_{10}^{m}\\ \begin{array}{c} C_{20}^{m}\\ \begin{array}{c} \ldots \\ C_{k0}^{m} \end{array} \end{array} \end{array} & \begin{array}{c} \begin{array}{cc} C_{11}^{m} & \begin{array}{cc} C_{12}^{m} & \begin{array}{cc} \ldots & C_{1r}^{m} \end{array} \end{array} \end{array}\\ \begin{array}{c} \begin{array}{cc} C_{21}^{m} & \begin{array}{cc} C_{22}^{m} & \begin{array}{cc} \ldots & C_{2r}^{m} \end{array} \end{array} \end{array}\\ \begin{array}{c} \begin{array}{cc} \ldots & \begin{array}{cc} \ldots & \begin{array}{cc} \ldots & \ldots \end{array} \end{array} \end{array}\\ \begin{array}{cc} C_{k1}^{m} & \begin{array}{cc} C_{k2}^{m} & \begin{array}{cc} \ldots & C_{kr}^{m} \end{array} \end{array} \end{array} \end{array} \end{array} \end{array} \end{array}\right|$$

When $L_{j}=N$, the chronicle is a model of the normal behavior of the considered system, otherwise ($L_{j}=f_{j})$ the chronicle is a model of the behavior of the system under the occurrence of the fault $f_{j}$. This methodology (CBAM) was proposed to address the problem of alarm management by developing reliable tools that support the analysis of event streams, in order to recognize activities that can generate normal or abnormal situations in complex flows [24,25]. The challenge is then to fit the formal recognition of behaviors into the context of Complex Event Processing. The dynamics of a process can be represented by an approach that depicts the behavior of the process using the events that occur during the process evolution. In this context, the chronicle approach [26] has been applied in many applications of situation recognition, often with a diagnosis objective. Chronicles are temporal patterns supported by a set of observable events and a set of temporal constraints between pairs of events [27]. One of the main difficulties of situation recognition based on chronicles is to obtain automatically a base of chronicles that represents each situation of interest. The proposal is then to use a chronicle recognition approach to analyze the behavior of the process, and to use learning techniques for the chronicles’ design. Diagnosis by situation recognition (chronicle-based diagnosis) in the startup and shutdown stages of mining/mineral/metal/chemical/petrochemical processes as a support for human operators is the principal goal of this new methodology, and it is resumed in the fact that super alarms can be generated according to the scenarios detected by the chronicles. In this paper, the hybrid system is represented by an extended transition system, whose discrete states represent the different modes of operation for which the continuous dynamics are characterized by a qualitative domain [28]. Formally, a hybrid causal system is defined as a tuple Γ = 〈V,D,Tr,E,*CSD*,Init,COMP, DCM〉, where:V = {*υ_i_*} is a set of continuous process variables which are functions of time.D is a set of discrete variables. D = Q⋃K⋃V_Q_, where:
○Q is a set of states qi of the transition system, which represents the system’s operation modes.○The set of auxiliary discrete variables K = {K_i_}, *I* = 1,2,3,….*n_c_* represents the system configuration in each mode *q_i_*, in which K_i_ indicates the discrete state of the active components.○V_Q_ is a set of qualitative variables whose values are obtained from the behavior of each continuous variable *υ_i_*.E = Σ ⋃ Σ^c^ is a finite set of observables (Σ*_o_*) and unobservable (Σ*_uo_*) event types, in which Σ is the set of event type associated to the procedural actions, for example, in the start-up or shutdown stages, and Σ^c^ is the set of event types associated to the behavior of the continuous process variables.Tr:Q × Σ → Q is the transition function. The transition from mode *q_i_* to mode *q_j_* with associated event *σ* is noted (*q_i_*,*σ*,*q_j_*).*CSD* ⊇ ⋃_i_*CSD_i_* is the Causal System Description or the causal model used to represent the constraints underlying the continuous dynamics of the hybrid system.

Every *CSD_i_* associated to a mode *q_i_*, is given by a graph *Gc* = V∪K, *I*, in which *I* is the set of influences in which there is an edge $\epsilon \left(\upsilon_{i},\upsilon_{j}\right)\in I$ from *υ_i_* ∈ V to *υ_j_* ∈ V if the variable *υ_i_* influences variable *υ_j_*. A dynamic control model $DCM_{I_{k}}$ is associated to every influence $I_{k}\in I$. Figure 5 presents the Dynamic Control Model where one procedural action *σ_i_* is related as an observable event that connects the industrial controller (PID) with the model of the active component (Comp. model) which corresponds to a transfer function of first order with delay. The event that closes the control loop *σ_j_* is assumed to be an unobservable event.

## 3. Results 

Oil transport is one important action in the petrochemical sector. The aim is to help the operator to recognize dangerous conditions during the start-up stage of an Oil Transport System, through the use of Super-Alarms. In this section, the petrochemical process analyzed is one unit of oil transport; see Figure 6. The measured continuous variables are the level *L* of the tank, the pressure *Po* in the pump, and the outlet flow *Qo(V2)* in valve V2. For the startup stage in this process, the initial conditions are that the tank (TK) is empty, the valves V1 and V2 are closed, and the pump Pu is off. In this situation, the alarms for the low levels in all of the continuous variables (L, Po and* Qo(V2)*) are active. For the shutdown stage in this process, the initial conditions could be different for each one of the others, depending on the situation in which the system is. For example, one condition is that the outlet pressure (Po) has passed its high limit activating the alarm PAH (Pressure Alarm High), but the outlet flow (*Qo(V2)*) does not increase over its low limit after that a specific quantity of time units has passed.

This Oil Transport System is composed of the following elements: sensors, passive components, and active components. The sensors are the level sensor (LT), the pressure sensor (PT), the inflow sensor (FT_1_) and the outflow sensor (FT_2_). The passive component is the tank (TK); in addition, the active components are two normally closed valves (V1 and V2), and one pump (Pu). Since there are three active components, the Oil Transport System obviously involves hybrid behavior. Modeling the behavior of this hybrid system involves a set of continuous variables and a set of discrete variables. The continuous variables are the level L, pressure Po, and outflow *Qo(V2)*, V = {L,Po,*Qo(V2)*}. The discrete variables are related to the operational actions of the process and the changes in the continuous variables, then the event types for this process are identified in the next sub-section.

### 3.1. Applying CBAM

In this subsection, the three steps of the Chronicle Based Alarm Management methodology are described.

#### 3.1.1. STEP 1: Event Type Identification

In the Oil Transport System of the case of this study, the set of event types Σ that represent the procedure actions is
(2)Σ={V1,V2,PuO, v1, v2,PuF, M2A}
where V1 (resp. V2) is for the action that switches the valve V1 (resp. V2) from closed to opened. On the other hand, v1 (v2) is the action that switches the valve V1 (resp. V2) from opened to closed, and PuO (resp. PuF) is for the action that turns on (resp. off) the pump. The event *M2A* corresponds to the transition from ‘manual’ to ‘automatic’ operation, closing the control loops. In the reminder of this discussion, we assume that this event is the unique unobservable event of the system, i.e., *M2A* ∈ Σ*_uo_*. The underlying DES (Discrete event system) of the Oil Transport System represents the sequence of observable procedure actions for a start-up stage (indicated by the red or green arrows in Figure 7, corresponding to the evolution of the operation modes (i.e., *q*_0_, *q*_1_, *q*_4_, *q*_5_, *q*_7_); for instance, in the mode of operation, *q*_1_ can be determined when the valve V1 is opened; therefore, the continuous variable *QiTK* influences the variable L, and the supervision system will wait for the event which indicates that after of a specific period of time, the level of water into the tank TK has passed its low limit. Each operation mode *q_i_* is associated with a causal system description to identify the influences between the variables L, Po and *Qo(V2)*. These influences allow the determination of the occurrence of the events Σ^c^.
(3)$$\Sigma^{c}=\left\{L_{\left(L\right)},l_{\left(L\right)},H_{\left(L\right)},h_{\left(L\right)},\;L_{\left(Po\right)},l_{\left(Po\right)},H_{\left(Po\right)},h_{\left(Po\right)},\;L_{\left(Qo\left(V2\right)\right)},l_{\left(Qo\left(V2\right)\right)},H_{\left(Qo\left(V2\right)\right)},h_{\left(Qo\left(V2\right)\right)}\right\}$$

*L*(L) indicates that the process variable L has passed its low level from down to up, and *l*(L) indicates that the process variable L has passed its low level from up to down. The same is true for the other variables *Po* and *Qo(V2)*.

#### 3.1.2. STEP 2: Event Sequence Generation

From simulations, the behavior of the variables is obtained, and the learning event sequences are generated according to the evolution of the system in each scenario. In this manuscript, the scenario of an abnormal start-up is analyzed. This abnormal situation is related to a failure in the valve V2. In this scenario, the sequences of the event types are similar to the event sequences of a normal start-up, until it is detected that the outlet flow in the system does not increase. When the level of oil in the tank TK arrived to its high limit, the ordered sequence of the event types that has occurred must be V1, L(L), H(L), PuO, V2 or V1, L(L), H(L), V2, PuO. In scenario 2a (V1, L(L), H(L), PuO, V2), the event type L(Po) occurs after V2. In scenario 2b (V1, L(L), H(L), V2, PuO), the event type L(Po) occurs after PuO. The event type H(Po) occurs after L(Po). Therefore, the ordered sequences of event types must be: V1, L(L), H(L), PuO, V2, L(Po), H(Po) or V1, L(L), H(L), V2, PuO, L(Po), H(Po). For this scenario, we chose the representative event sequences (S_1_, S_2_ and S_3_) that show the extreme behaviors of all of the possible sequence orders of the event types.S_1_ = 〈(V1,1); (L(L),21); (H(L),48); (PuO,50); (V2,51); (L(Po),60); (H(Po),75)〉S_2_ = 〈(V1,1); (L(L),25); (H(L),55); (V2,56); (PuO,57); (L((Po),63); (H(Po),78)〉S_3_ = 〈(V1,1); (L(L),28); (H(L),60); (PuO,61); (V2,62); (L(Po),71); (H(Po),85)〉

The simulation of this abnormal start-up is presented in Figure 8, where the evolution of the variables *L* and *Po* is represented. The variable *Qo(V2)* does not appear, because the valve V2 has failed. The values of the variables are specified as follows:For the variable of the level (*L*), the value of 0 corresponds to 0 m; each increase of 2 (vertical axis) corresponds to 2 m.For the variable of the pressure (*Po*), the value of 0 corresponds to 0 PSI; each increase of 2 (vertical axis) corresponds to 20 psi.

#### 3.1.3. STEP 3: Chronicle Database Construction

This chronicle database is to be submitted to a chronicle recognition system that identifies in an observable flow of events, all of the possible matching with the set of chronicles. Chronicles from which the situation (normal or faulty) can be assessed by generating a Super-Alarm. The chronicle *C*^1^_11_ from the set of chronicles of the Oil Transport System is presented, i.e., of the area *Ar*_1_ of the whole system. Therefore, the chronicle *C*^1^_11_ is associated with a failure behavior of type *f*_1_ during a start-up stage. In the figures of the chronicles, the events are specified as follows: L(L) as LL; l(L) as lL; H(L) as HL; h(L) as hL; L(Po) as LP; L(Po) as lP; H(Po) as HP; h(Po) as hP; L(Qo(V 2)) as LQ; l(Qo(V 2)) as lQ; H(Qo(V 2)) as HQ; h(Qo(V 2)) as hQ. For the scenario of an abnormal start-up, the following temporal restrictions are used in the extended version of the HCDAM (Heuristic Chronicle Discovery Algorithm) [23]. The notation TR_PuO,V2_ = PuO[−2,2]V2 corresponds to a temporal restriction which indicates that the valve V2 can be opened (V2) two time units before that the pump Pu is turned on (PuO) or, on the contrary, that PuO occurs two time units before that of V2. On the other hand, the temporal restriction noted as TR_HL,PuO_ = HL[1,4]PuO, expresses that the pump Pu is turned on (PuO) between one and four time units after that the high limit level into the tank happens (HL). The chronicle *C*^1^_11_ that resulted using the algorithm HCDAM is presented in Figure 9. The learning event sequences used are the S_1_, S_2_ and S_3_ which were generated before (STEP 2).

### 3.2. Validation

This section presents the evaluation of the chronicle *C*^1^_11,_ which represents the temporal pattern for an abnormal start-up in the Oil Transport System. One sequence of evaluation that belongs to this abnormal scenario is presented below: S_eval_ = ⟨(V1,1);(LL,26);(HL,58);(PuO,60);(V2,62);(LP,70);(HP,85)⟩, which is different to the learning event sequences, and it expresses an abnormal condition of start-up. Figure 10, Figure 11, Figure 12, Figure 13, Figure 14, Figure 15 and Figure 16 present the recognition process of the chronicle and the generation of one Super-Alarm. In Figure 10, the first occurrence is (V1, 1); the next occurrence must be of the event LL between 20 and 28 time-units. Now, in Figure 11, the activation of LL at 26 is presented, indicating also that the next occurrence must be HL. The following events occur (PuO, V2, LP and HP) until the chronicle is recognized and the super alarm is generated. Therefore, this new element (the Super-Alarm) corresponds to one superior alarm that gives the relevant information to the operators after a diagnosis process, increasing the reliability of this protective layer.

## 4. Conclusions

A new layer of protection in industrial processes has been proposed. This new layer is called a Super-Alarm, which refers to a new alert to the operators resulting from a diagnosis procedure representing a superior alarm. Furthermore, a new methodology for the alarm management of complex processes has been proposed, in order to generate Super-Alarms. This methodology proposes a diagnosis process as a support tool to the operators during transitional stages, based on situation recognition. The situations to recognize correspond to the normal and/or abnormal process behaviors modeled by temporal patterns called Chronicles. The case study illustrates the construction of a chronicle of an abnormal start-up of an oil transport system, and then shows the way how a Super-Alarm is generated. Any additional protection layer that increases the reliability of the industrial processes is well received, because the risk of accidents and failures in which human lives are involved can be reduced. Therefore, this proposal could increase the number of tools and components that help the operators to detect early hazard situations, and the risk analysis methods such as fault trees, bow ties, etc. can be used to construct models of failure scenarios in a supervision system. The future work will be related to the implementation of this new concept in the supervision tools of an industrial process (energy, chemical, mining), and will use V-nets [29], guaranteeing the reliability of the diagnosis tool.

## Figures and Tables

**Figure 1 entropy-23-00139-f001:**
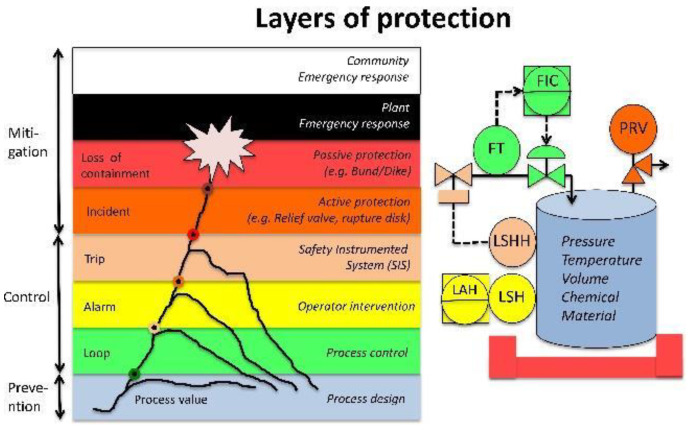
Safety layers of protection.

**Figure 2 entropy-23-00139-f002:**
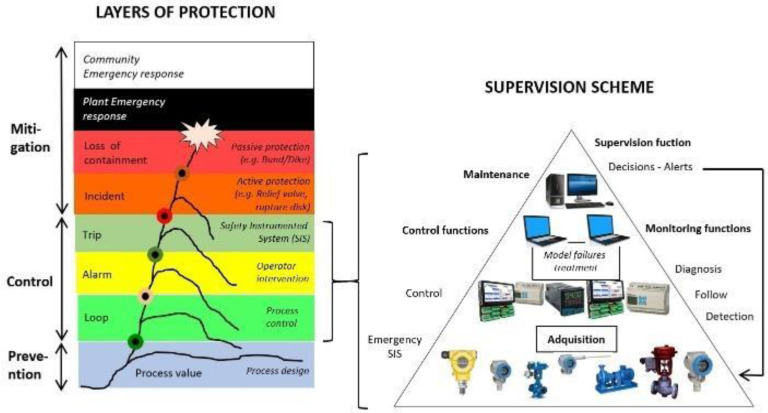
Process safety relationships.

**Figure 3 entropy-23-00139-f003:**
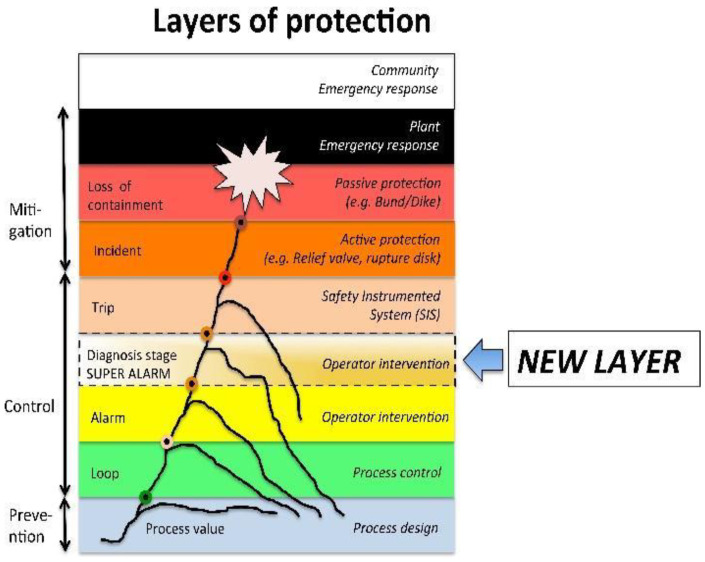
New layer of protection called a Super-Alarm.

**Figure 4 entropy-23-00139-f004:**
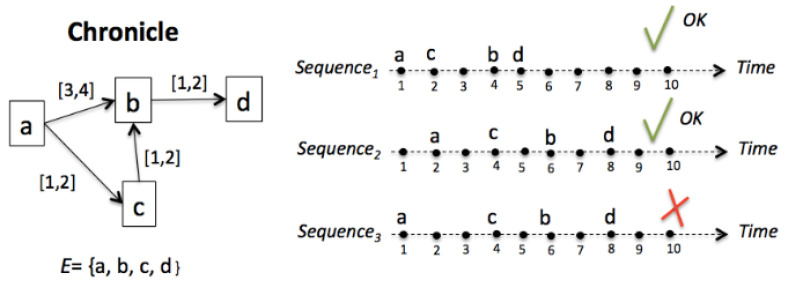
Chronicle instance.

**Figure 5 entropy-23-00139-f005:**
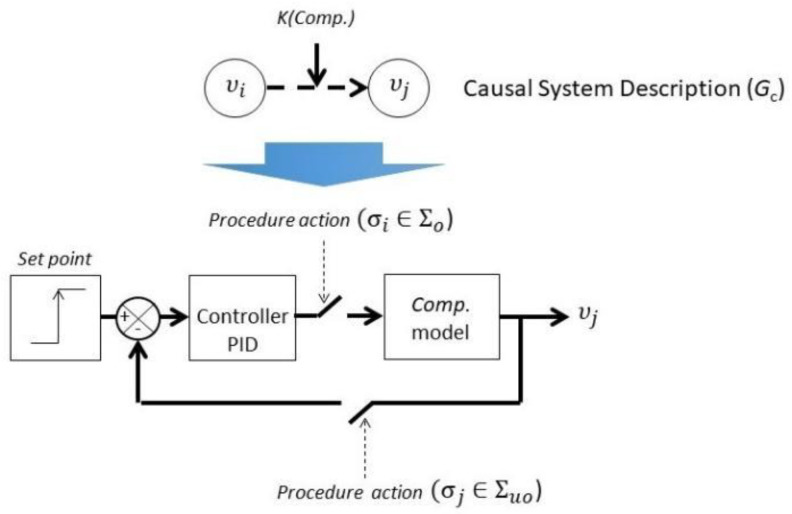
Dynamic Control Model (DCM).

**Figure 6 entropy-23-00139-f006:**
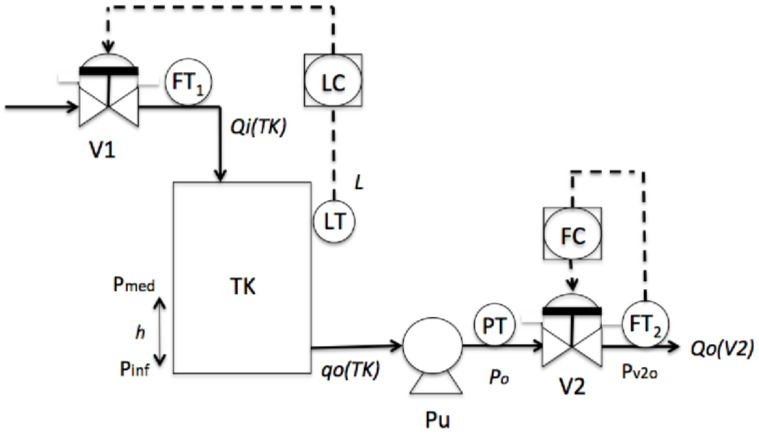
Oil Transport System unit.

**Figure 7 entropy-23-00139-f007:**
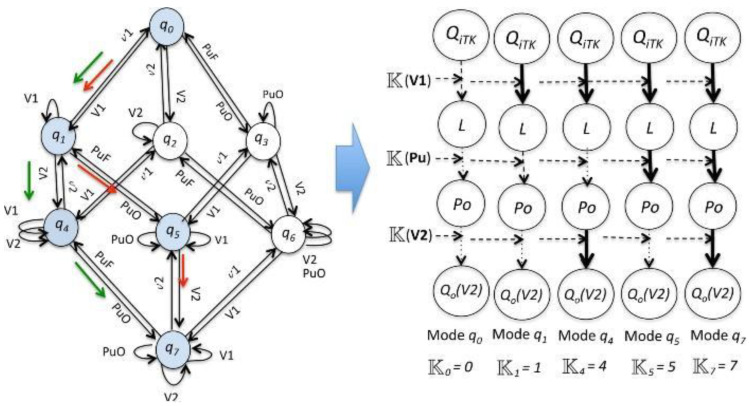
Start-up stage of the Oil Transport System: the underlying DES and Causal System Description.

**Figure 8 entropy-23-00139-f008:**
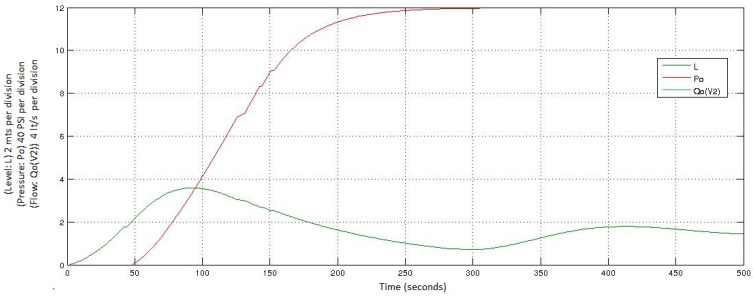
Simulation of a startup with a failure in V2 in the Oil Transport System.

**Figure 9 entropy-23-00139-f009:**
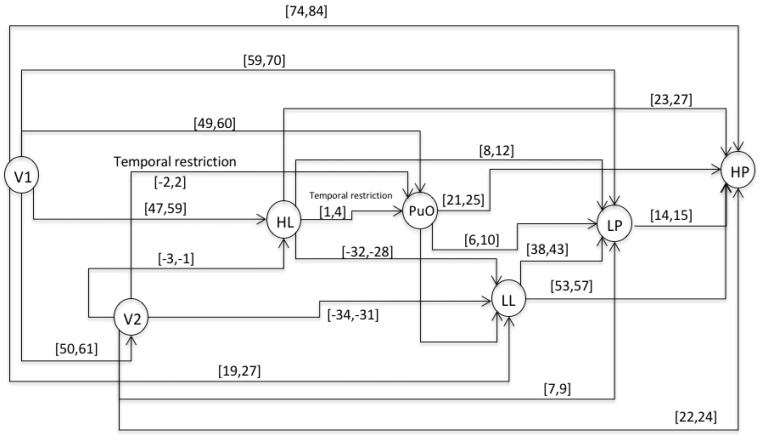
Directed graph (*G*) of the chronicle *C*^1^_11_.

**Figure 10 entropy-23-00139-f010:**
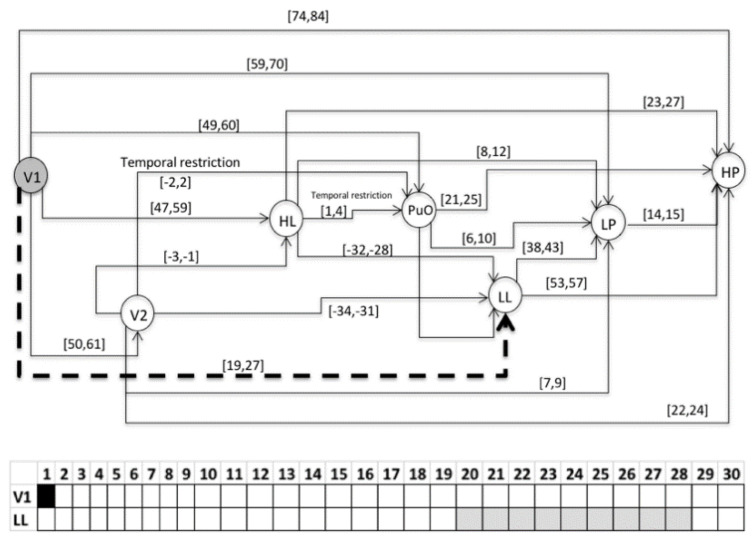
Activation of V1 at 1.

**Figure 11 entropy-23-00139-f011:**
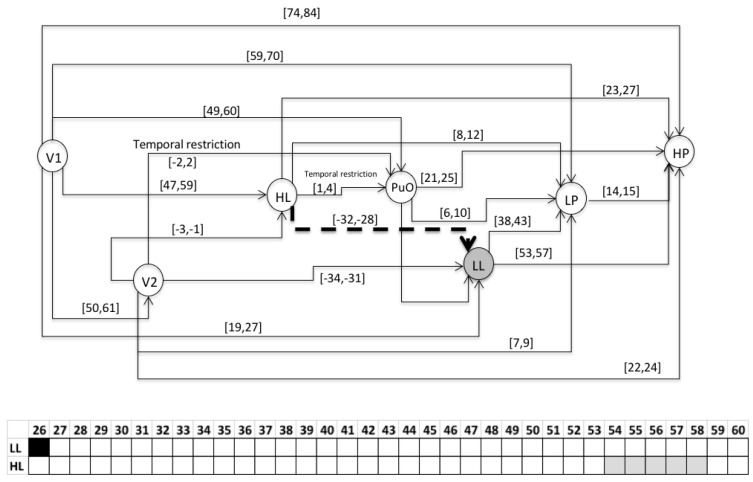
Activation of LL at 26.

**Figure 12 entropy-23-00139-f012:**
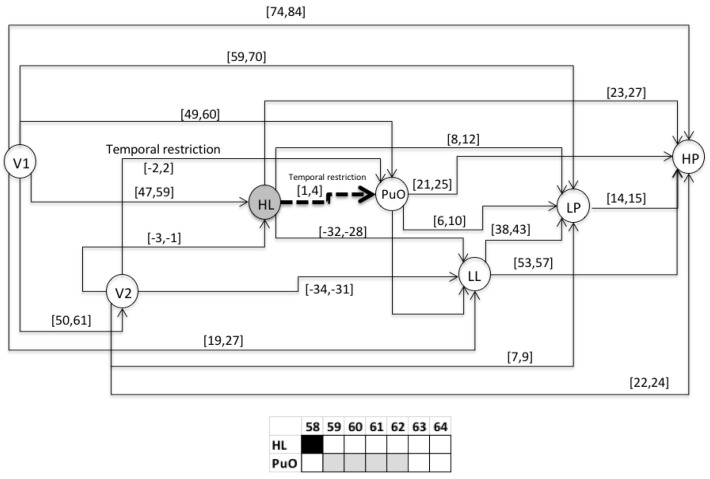
Activation of HL at 58.

**Figure 13 entropy-23-00139-f013:**
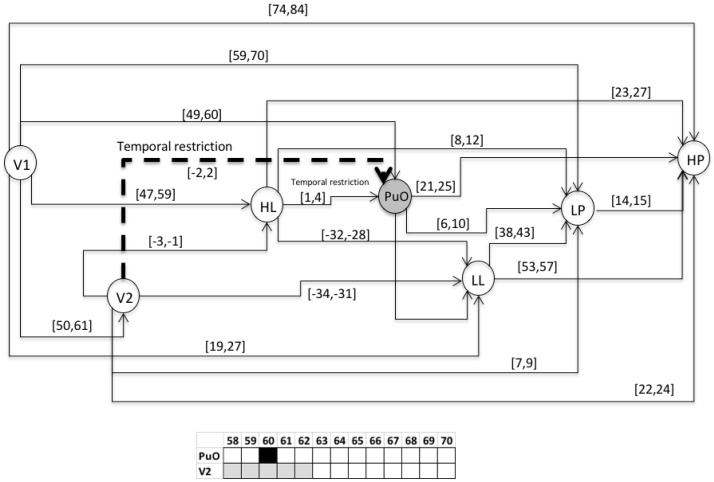
Activation of PuO at 60.

**Figure 14 entropy-23-00139-f014:**
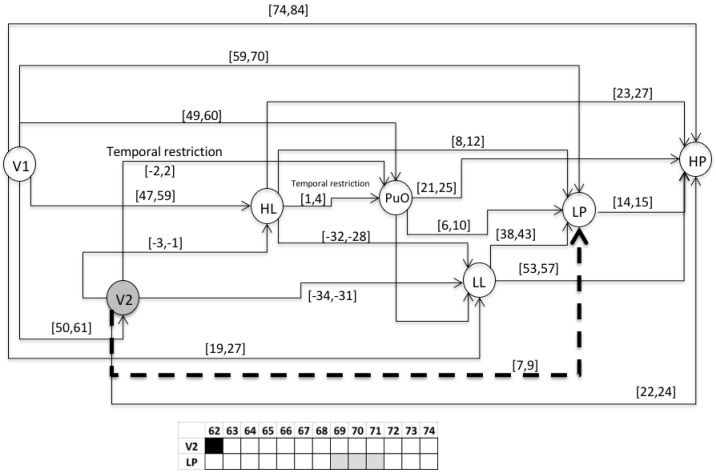
Activation of V2 at 62.

**Figure 15 entropy-23-00139-f015:**
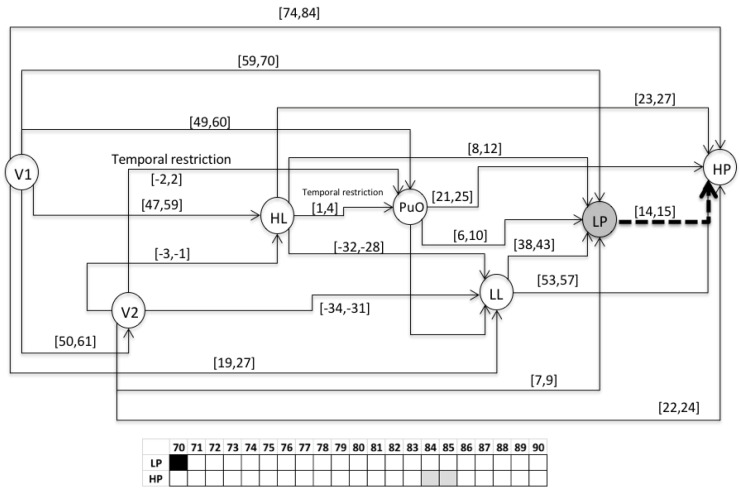
Activation of LP at 70.

**Figure 16 entropy-23-00139-f016:**
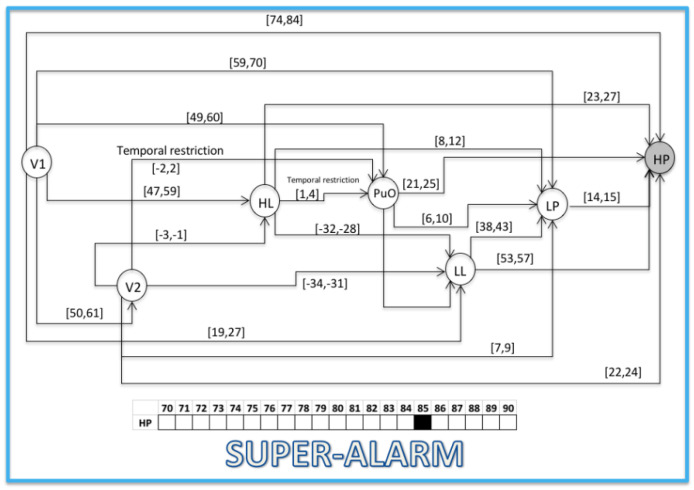
Activation of HP at 85; the abnormal situation recognized, generating a Super-Alarm.

## Data Availability

The data presented in this study is available on request from the corresponding author.

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
