# Peer review of "Super-Alarms with Diagnosis Proficiency Used as an Additional Layer of Protection Applied to an Oil Transport System"

_entropy, 2021, doi:10.3390/e23020139_

Round 1
Reviewer 1 Report
1.
Even though the topic may be of interest, the paper is hard to read: punctuation and grammar must be improved, and there are sentences to be rephrased. The corrections are annotated in the attached file. I recommend a further check before resubmission.
2.
The introduction (sec. 1) consists of one long paragraph. I suggest the organization of sec. 1 in two or more paragraphs.
3.
Fig. 14 and its caption are in different pages. The caption should be close to the figure. In fig. 10-16 I suggest the replacement of "at 1" with "at time 1", the replacement of "at 26" with "at time 26", and so on.

Author Response
Dear reviewer
Thanks for your comments and suggestions. We adjust all the paper following your indications. Please see the attachment in which you will find a new version of the article.
Best regards
John William Vásquez Capacho PhD

Reviewer 2 Report
Consequential alarms are common in complex industrial plants. However, easy implementation of alarms may lead to overwhelming number of alarms, resulting in the low efficiency of alarm management. In layers of protection, alarms are used to remind operators of handling abnormalities. So advanced alarms could provide more meaningful indications in addition to the original alarms. In this paper, super alarms are defined according to the temporal features of alarms to realize the so-called Chronicles Based Alarm Management (CBAM). This is an interesting idea.
There are some related works that imply the idea of super alarms:
Online pattern matching and prediction of incoming alarm floods. Journal of Process Control, 2017
An accelerated alignment method for analyzing time sequences of industrial alarm floods. Journal of Process Control, 2017
Accelerated multiple alarm flood sequence alignment for abnormality pattern mining. Journal of Process Control, 2019
There are also review papers about alarm management:
An overview of industrial alarm systems: Main causes for alarm overloading, research status, and open problems. IEEE Transactions on Automation Science and Engineering, 2016
The above paper also mentions the idea of alarm pattern matching based on temporal information.
I noticed a similar paper by the same authors:
An additional layer of protection through superalarms with diagnosis capability. CT&F - Ciencia, Tecnologia y Futuro
Should it be cited in this paper?
My major concerns are:
The propagation time from one alarm to another may change due to disturbance and uncertainty. In some cases, the order of alarms may even reversed. Is the proposed method tolerate such cases?
In real-time applications, how about the performance? Because we could not wait for the whole sequence but have to do analysis as early as possible.
How to build the chronicles?
Author Response
Dear reviewer,
Thanks for your comments. Please see the attachment
Best regards
John William Vásquez Capacho PhD

Round 2
Reviewer 2 Report
The recent paper by the same authors should be cited, as mentioned in my previous review.